# Cloning and Characterization of Yak *DHODH* Gene and Its Functional Studies in a Bisphenol S-Induced Ferroptosis Model of Fetal Fibroblasts

**DOI:** 10.3390/ani13243832

**Published:** 2023-12-13

**Authors:** Hongmei Xu, Yueyue Li, Qiao Li, Zifeng Ma, Shi Yin, Honghong He, Yan Xiong, Xianrong Xiong, Daoliang Lan, Jian Li, Wei Fu

**Affiliations:** 1Key Laboratory of Animal Science of National Ethnic Affairs Commission of China, Southwest Minzu University, Chengdu 610041, China; hongmeixu99@163.com (H.X.); liyueyue0214@163.com (Y.L.); li17726564323@163.com (Q.L.); mzifeng2023@163.com (Z.M.); raulyinshi@163.com (S.Y.); honghong3h@126.com (H.H.); xiongyan0910@126.com (Y.X.); 22100093@swun.edu.cn (X.X.); landaoliang@163.com (D.L.); 2Key Laboratory of Qinghai-Tibetan Plateau Animal Genetic Resource Reservation and Utilization, Ministry of Education, Southwest Minzu University, Chengdu 610041, China; 3Key Laboratory of Qinghai-Tibetan Plateau Animal Genetic Resource Reservation and Utilization, Sichuan Province, Southwest Minzu University, Chengdu 610041, China

**Keywords:** yak, *DHODH* gene, fetal fibroblasts, ferroptosis, Bisphenol S

## Abstract

**Simple Summary:**

In recent years, authoritative journals have consistently reported DHODH’s critical role in resistance to ferroptosis, significantly advancing the research on related mechanisms. Nevertheless, the nucleotide sequence of the yak *DHODH* gene remains unknown, and its involvements in ferroptosis processes in yak cells remain elusive. Therefore, in the present study, we cloned the coding region of the yak *DHODH* gene, and revealed its high conservation in mammals. Furthermore, by using a bisphenol S (BPS)-induced ferroptosis model, we confirmed the role of the *DHODH* gene in mitigating ferroptosis in yak skin fibroblasts (YSFs) derived from a three-mouth-old fetus. This study offers theoretical support for further exploring the functions of the yak *DHODH* gene and partially elucidating the molecular mechanisms underlying ferroptosis.

**Abstract:**

Dihydroorotate dehydrogenase (DHODH) is a rate-limiting enzyme of *de novo* biosynthesis of pyrimidine. Although the involvement of DHODH in resisting ferroptosis has been successively reported in recent years, which greatly advanced the understanding of the mechanism of programmed cell death (PCD), the genetic sequence of the yak *DHODH* gene and its roles in ferroptosis are still unknown. For this purpose, we firstly cloned the coding region sequence of *DHODH* (1188 bp) from yak liver and conducted a characterization analysis of its predictive protein that consists of 395 amino acids. We found that the coding region of the yak *DHODH* gene presented high conservation among species. Second, the expression profile of the *DHODH* gene in various yak tissues was investigated using RT-qPCR. The results demonstrated that *DHODH* was widely expressed in different yak tissues, with particularly high levels in the spleen, heart, and liver. Third, to investigate the involvement of *DHODH* in regulating ferroptosis in cells, yak skin fibroblasts (YSFs) were isolated from fetuses. And then, bisphenol S (BPS) was used to induce the *in vitro* ferroptosis model of YSFs. We observed that BPS decreased the cell viability (CCK8) and membrane potential (JC-1) of YSFs in a dose-dependent manner and induced oxidative stress by elevating reactive oxygen species (ROS). Simultaneously, it was evident that BPS effectively augmented the indicators associated with ferroptosis (MDA and BODIPY staining) and reduced GSH levels. Importantly, the co-administration of Ferrostatin-1 (Fer), a potent inhibitor of ferroptosis, significantly alleviated the aforementioned markers, thereby confirming the successful induction of ferroptosis in YSFs by BPS. Finally, overexpression plasmids and siRNAs of the yak *DHODH* gene were designed and transfected respectively into BPS-cultured YSFs to modulate *DHODH* expression. The findings revealed that *DHODH* overexpression alleviated the occurrence of BPS-induced ferroptosis, while interference of *DHODH* intensified the ferroptosis process in YSFs. In summary, we successfully cloned the coding region of the yak *DHODH* gene, demonstrating its remarkable conservation across species. Moreover, using BPS-induced ferroptosis in YSFs as the model, the study confirmed the role of the *DHODH* gene in resisting ferroptosis in yaks. These results offer valuable theoretical foundations for future investigations into the functionality of the yak *DHODH* gene and the underlying mechanisms of ferroptosis in this species.

## 1. Introduction

The highland ecosystem, as a prime example of biological adaptability and survival mechanisms in harsh environments, has persistently commanded the profound interest of ecologists and biologists [1]. In the frigid alpine ecosystem, the yak (*Bos grunniens*), emblematic of the highland terrain, confronts the trials of survival, braving extreme temperatures, scarce oxygen, and restricted food availability [2,3]. The biological mechanisms underlying yak growth, reproduction, and survival are highly vulnerable to disruption caused by various external stressors, thereby constraining their population size and reproductive capabilities [4,5]. Yaks possess exceptional evolutionary adaptations, enabling them to survive in alpine environments, and the molecular pathways orchestrating their response to harsh conditions remain complex and not yet fully elucidated [6].

DHODH is a key part of the cellular purine biosynthetic pathway [7,8]. specifically, it is involved in the fourth step of *de novo* pyrimidine synthesis, where it converts dihydroorotate acid into orotate. Subsequently, orotate undergoes two essential steps to form the precursor of pyrimidine nucleotides, namely uridine monophosphate (UMP) [9,10]. Purine nucleotides serve as pivotal players in the cellular biosynthesis of DNA and RNA, serving as indispensable constituents of adenosine triphosphate (ATP) as well [11,12]. The enzymatic activity of DHODH is very important in cellular metabolic balances, being a pivotal part in the intricate processes of growth and development [13]. Furthermore, as a key constituent of the mitochondrial respiratory chain, DHODH carries out an essential function in maintaining cellular metabolic equilibrium and ensuring optimal energy supply [14,15]. Studies have indicated that DHODH has a substantial impact on the regulation of programmed cell death (PCD), ferroptosis for instance, and shows extraordinary effects on the maintenance of mitochondrial membrane potential, regulation of redox balance, and inhibition of oxidative stress response [16]. Furthermore, DHODH demonstrates a distinct mechanism for counteracting ferroptosis when compared to key players in the inhibition of ferroptosis, such as glutathione peroxidase 4 (GPX4) and apoptosis-inducing factor mitochondria-associated 2 (AIFM2). GPX4 primarily exerts its anti-ferroptosis effects by preventing lipid peroxidation, whereas the role of AIFM2 lies in maintaining mitochondrial membrane potential and overall cellular functions [17,18]. Nevertheless, DHODH exerts its impact on the cellular milieu by regulating pyrimidine nucleotides, thereby influencing the occurrence of ferroptosis [19]. Although the roles of DHODH in various species have been gradually elucidated, the exploration of its functions in yaks remains unexplored. Consequently, the cloning of the yak *DHODH* gene coding sequence would contribute to a more comprehensive understanding of its potential roles in yak physiological adaptation and stress response. This study also enables a deeper investigation into yak survival strategies in challenging high plateau environments.

Currently, environmental estrogens (EEs) have emerged as an omnipresent element of the ecological environment, inflicting widespread and significantly detrimental effects on the health and survival of both human beings and animals. EEs possess the potential to disrupt the normal functioning of the endocrine system, thereby leading to adverse impacts on the reproductive and developmental processes of organisms [20]. This disruption may lead to decreased reproductive ability, increased fertility issues, and even pose a potential threat to the long-term survival of species [21]. Due to its horrible perturbation on the reproductive endocrine system, Bisphenol A (BPA), a typical environmental estrogen, has been banned in the production of baby products in the majority of countries worldwide [22]. However, Bisphenol S (BPS), an alternative to BPA, has gained widespread use in various consumer goods on account of its perceived low toxicity [23]. Nevertheless, recent research has revealed that BPS can still elicit toxic effects on organisms under certain conditions, particularly by altering markers of ferroptosis, such as lipid peroxidation [24]. Although some studies have suggested that BPA can induce ferroptosis in mouse testes and renal tubular epithelial cells [25,26], the underlying molecular mechanisms of ferroptosis associated with BPS remain elusive.

The objective of this study was to explore the fundamental role of the *DHODH* gene in governing ferroptosis in yaks. By conducting the first-ever cloning and sequencing of the coding region of the yak *DHODH* gene, we unveiled its distinctive characteristics and potential biological functions specific to this species. Next, RT-qPCR was employed to generate the expression profile of the *DHODH* gene from various yak tissues. Subsequently, the primary cell line of YSFs was isolated to explore the potential role of the *DHODH* gene in regulating ferroptosis in yak cells, which involved establishing a BPS-induced ferroptosis model in YSFs and validating it via co-treatment with the ferroptosis inhibitor Ferrostatin-1. Moreover, overexpression vectors and siRNA targeting *DHODH* were designed and transfected into BPS-cultured fibroblasts to further elucidate the underlying mechanism of *DHODH* in regulating ferroptosis in YSFs. This study aims to elucidate the contribution of *DHODH* to the ferroptosis process in yak cells, offering novel insights into the mechanism by which plateau organisms respond to challenging environmental conditions.

## 2. Materials and Methods

The animal experiments were checked and approved by the Ethics Committee of Southwest Minzu University (the Ethic Approval code: SMU-202309002), and performed in accordance with the Animal Care and Use Program Guidelines of the Sichuan Province, China.

### 2.1. Sample Collection

Samples of yak heart, liver, spleen, lungs, kidneys, testes, epididymis, ovaries, uterus, placenta, muscles, and fat tissues were obtained from a slaughterhouse in Hongyuan County, Sichuan Province, China. The tissues were immediately collected and then rinsed with PBS, diced into small pieces, and stored in cryovials immersed in liquid nitrogen. Additionally, yak fetuses at approximately three months of gestation were obtained along with the uterus and soaked in pre-cooled saline. The fetal skin tissues were collected within six hours after transportation to the laboratory.

### 2.2. Gene Cloning and Construction of the Overexpression Vector of Yak DHODH Gene

The coding region of the yak *DHODH* gene was amplified from cDNA of yak liver using polymerase chain reaction (PCR). The PCR amplification consisted of 50 μL, comprising 2 μL of cDNA template, 1 μL each of forward and reversed primers, 25 μL of 2× Fast Taq PreMix (Vazyme, P01-7, Nanjing, Jiangsu, China), and 21 μL of ddH_2_O. The PCR program was carried out as follows: pre-denaturation at 98 °C for 30 s, followed by denaturation at 98 °C for 10 s, annealing at 60 °C for 20 s, and extension at 72 °C for 30 s (repeated for 35 cycles from denaturation to extension), and finally, a final extension step was performed at 72 °C for 5 min. The amplification product was resolved on a 1.0% agarose gel, and subsequently, the target DNA was purified using a DNA gel extraction kit (Vazyme, DC301, Nanjing, Jiangsu, China). The fragments obtained were directionally cloned into pcDNA3.1(+) vector (Thermo Fisher, V79020, Waltham, MT, USA) and then transfected into competent DH5α cells (Vazyme, C502-02, Nanjing, Jiangsu, China). Positive clones were confirmed by Sanger sequencing conducted by Tsingke Biotechnology Co., Ltd. (Tsingke, Beijing, China). Moreover, PCR was used to isolate the coding region (CDS) of *DHODH* gene from yak liver, along with the Xhol and EcoRI homologous sequences on pcDNA3.1(+) vector for subsequent recombination. The primers used in this process were synthesized by TsingKe, and the primer sequences can be found in Appendix A.

### 2.3. Bioinformatics Analysis of the Coding Region of Yak DHODH Gene

The prediction of the coding region of the yak *DHODH* gene was conducted using the tool NCBI ORF Finder (https://www.ncbi.nlm.nih.gov/orffinder/, accessed on 15 June 2023). Homology comparison of nucleotide and predicted amino acid was performed using DNAMAN. Various biological analyses were carried out using the following tools: ProtParam (https://www.expasy.org/search/Protparam, accessed on 15 June 2023), ProtScale (https://www.expasy.org/resources/protscale, accessed on 16 June 2023), NetPhos 3.1 (NetPhos 3.1—DTU Health Tech—Bioinformatic Services), NetNGlyc 1.0 (https://services.healthtech.dtu.dk/services/NetNGlyc-1.0/, accessed on 18 June 2023), TMHMM-2.0 (https://services.healthtech.dtu.dk/service.php?TMHMM-2.0, accessed on 18 June 2023), GenScript (https://www.genscript.com/wolf-psort.htm, accessed on 19 June 2023), SignalP 6.0 (https://services.healthtech.dtu.dk/service.php?SignalP, accessed on 22 June 2023), WoLF PSORT (https://wolfpsort.hgc.jp/, accessed on 25 June 2023), and SWISS-MODEL (https://swissmodel.expasy.org/interactive, accessed on 30 June 2023).

### 2.4. Total RNA Extraction and Quantitative Real-Time PCR (RT-qPCR)

The yak tissues were ground into a fine powder using a pre-chilled mortar and pestle. A 0.1 g amount of tissue powder was collected and transferred into 1.5 mL centrifuge tube. The total RNA was extracted using a BioSharp RNA extraction kit (Biosharp, BS258A, Hefei, Anhui, China). The concentration of RNA was determined using a UV-visible spectrophotometer (Shimadzu, Biospec-nano, Shimadzu, Japan) at the wavelength of 260 nm. Afterwards, 1 μg of total RNA was reverse-transcribed into complementary DNA (cDNA) using the RNA reverse transcription kit (Vazyme, M1631, Nanjing, Jiangsu, China), which was used as the template in the subsequent RT-qPCR analysis. The RT-qPCR analysis was performed to determine the expression profile of the *DHODH* gene with a total reaction volume of 10 μL, consisting of 0.5 μL each of forward and reverse primers, 1 μL of cDNA, 5 μL of 2× ChamQ Universal SYBR qPCR Master Mix (Vazyme, R223, Nanjing, Jiangsu, China), and 3 μL of ddH_2_O. The *GAPDH* gene was used as the reference, and the relative expression of the *DHODH* gene was calculated using the 2^−ΔΔCt^ method. The primer sequences can be found in Appendix A.

### 2.5. Isolation and Culture of Yak Fetal Fibroblasts (YSFs)

In this study, a set of rigorous protocols were implemented to ensure the activation and high quality of YSFs. Initially, the entire fetus-uterus tissues were immersed in 70% ethanol for 1 min. Subsequently, the fetus was isolated and placed in pre-cooled PBS. Next, appropriate skin samples were taken and thoroughly rinsed with PBS containing 1% penicillin/streptomycin (Gibco, C20012500BT, Shanghai, China) in order to remove any external contaminants. The tissue was then transferred to a sterile environment and finely cut into approximately 1 mm^3^ cubes. To minimize mechanical damage and maintain cell viability, the skin tissues were immersed in cell culture medium (DMEM/F12 medium containing 10% fetal bovine serum (Newzeru, FBS-NZ500, Christchurch, New Zealand) and 2% P/S) throughout the processing steps. Subsequently, the tissue was minced completely, and 0.25% trypsin (Hyclone, SH30042.02, Logan, UT, USA) was added to disperse cells. The mixture was then incubated in a 5% CO_2_ humidified environment at 37 °C for 15 min. Following this, collagenase I (Solarbio, C8140, Beijing, China) was introduced into the tubes at a final concentration of 0.5 mg/mL to further release the cells for 1 h. After centrifugation, the supernatant containing the enzymes was removed, and the dispersed cells were resuspended in the cell culture medium. Finally, the suspended cells were cultivated onto 6 mm dishes and cultured in a 5% CO_2_ incubator at 37 °C. The cell culture medium was refreshed daily. Once the confluency reached 90%, the cells were trypsinized and passaged at a ratio of 1:3.

### 2.6. Cell Counting Kit-8 Assay

YSFs were seeded at a density of 7 × 10^3^ cells per well in a 96-well plate. Cells received bisphenol S (Aladdin, 80-09-1, Shanghai, China) treatment alone or in combination with vector/siRNA transfection. Following 12 h, 24 h, and 48 h after treatment, 10 μL of Cell Counting Kit-8 (Biosharp, BS350A, Anhui, China) was added in each well and incubated in a humidifier for 2.5 h. Subsequently, absorbance at 450 nm was measured by using a microplate reader (Thermo Scientific, Multiskan Sky, Waltham, MA, USA). The cell viability was calculated as follows: cell viability = [OD (treatment) − OD (blank)]/[OD (control) − OD (blank)] × 100%.

### 2.7. Reactive Oxygen Species (ROS) Detection

The intracellular level of reactive oxygen species (ROS) was determined by measuring the fluorescence intensity of intracellular DCF using the Reactive Oxygen Species Assay Kit (Biosharp, BL714A, Hefei, Anhui, China). YSFs were plated at a density of 6 × 10^5^ cells per well in a 12-well plate. After vector/siRNA transfection for 24 h, cells were treated with BPS. Then, the treatment medium was removed, and 500 μL of working solution containing 10 μM H_2_DCFDA was added to each well. The cells were incubated at 37 °C in a humidified environment for 30 min. After washing the cells twice with serum-free culture medium, a laser confocal microscope (ZEISS, LSM800, Oberkochen, Germany) was used to observe the cells and capture photographs. All procedures were conducted in the dark.

### 2.8. JC-1 Mitochondrial Membrane Potential Detection

Mitochondrial membrane potential was assessed using the JC-1 mitochondrial membrane potential detection kit (JC-1) (UElandy, J6004s, Suzhou, Jiangsu, China). The cell preparation was identical to that employed for ROS detection, and the experimental procedures were carried out following the kit’s instructions. In brief, a mixture of 5 μL of 100× JC-1 staining solution and 222.5 μL of ddH_2_O was prepared. Next, 25 μL of 10× Assay Buffer and 250 μL of 1× DMEM basic culture medium (Gibco, C11330500BT, Grand Island, NY, USA) were added to the mixture to obtain a final volume of 500 μL for JC-1 staining working solution. Afterward, the culture medium was removed, and the cells were washed twice with DPBS. Subsequently, 500 μL of JC-1 staining working solution was added to each well, and the plate was incubated in a humidified environment for 20 min. After washing twice with DPBS, 1 mL of serum-free DMEM culture medium was added to each well. Finally, the cells were observed and imaged using a laser confocal microscope (ZEISS, LSM800, Oberkochen, Germany).

### 2.9. C11-BODIPY581/591 Staining

C11-BODIPY581/591 (Cayman Chemical, 27086, Ann Arbor, MI, USA) was used for assessing cellular oxidative stress and lipid oxidation levels. The cell preparation remained the same as that for ROS detection. Initially, the BODIPY staining working solution was prepared by mixing 498 μL of 1× DMEM basic culture medium, 1 μL of a 1 mM BODIPY solution, and 1 μL of a 1 mM Hoechst solution, resulting in a final volume of 500 μL. The culture medium was then aspirated, and the cells were washed twice with DPBS. Subsequently, the plate was incubated in a humidified environment for 30 min. After washing twice with DPBS, 1 mL of serum-free DMEM culture medium was added to each well. Finally, the cells were observed and photographed using a laser confocal microscope (ZEISS, LSM800, Oberkochen, Germany).

### 2.10. Reduced Glutathione (GSH) Content Assay

The Reduced Glutathione (GSH) Content Assay Kit (Biosharp, BL874B, Hefei, Anhui, China) was used to assess cellular oxidative stress levels. And the cell preparation method was identical to that for ROS detection. The cells were washed twice with DPBS after removing the culture medium. Next, 200 μL of extraction solution was added to each well. The cells were scraped from the dishes using a BeyoGold™ 21 cm cell scraper (Beyotime, FLFT021, Shanghai, China) and collected in a 1.5 mL centrifuge tube. To lyse the cells, the tubes were alternated between liquid nitrogen and a 37 °C water bath 25 times. Afterward, the lysates were centrifuged at 12,000 rpm at 4 °C for 15 min, and the resultant supernatants were collected for analysis. In order to measure the GSH content in the samples, 20 μL of the supernatant, 120 μL of Reagent I, and 40 μL of Reagent II were sequentially added to a 96-well plate. After thorough mixing, the plate was incubated at room temperature in the dark for 5 min. The microplate reader (Thermo Scientific, Multiskan Sky, Waltham, MA, USA) was used to measure the absorbance at 412 nm (ΔA). Additionally, protein quantification was carried out by using a BCA protein concentration determination kit (Enhanced) (Beytime, P0010, Shanghai, China) (Cpr). The GSH content in μmol/mg was calculated according to the formula: GSH = [(ΔA + 0.0037)/0.5834]/Cpr.

### 2.11. Lipid Peroxidation MDA Assay

The cellular levels of lipid oxidation were determined using the Lipid Peroxidation MDA Assay Kit (Biosharp, BL904A, Hefei, Anhui, China). Sample collection and lysis procedures were conducted in the same manner as the GSH assay. The following procedure was followed for the preparation of MDA detection working solution: First, 150 μL of TBA dilution solution, followed by 50 μL of TBA storage solution, and finally, 3 μL of antioxidant were sequentially added and thoroughly mixed for a single sample. After that, 200 μL of the MDA detection working solution was added to 100 μL of the sample supernatant. The resulting mixture was then heated in a metal bath at 100 °C for 15 min. After the samples were cooled to room temperature, they were centrifuged at 1000 rpm for 10 min. Following centrifugation, 200 μL of the supernatant was transferred to a 96-well plate, and the absorbance was measured at 532 nm using a microplate reader (Thermo Scientific, Multiskan Sky, Waltham, MA, USA) (ΔA). In addition, the protein concentration was determined using the BCA protein concentration determination kit (Beyotime, P0012, Shanghai, China) (Cpr). The MDA content in μmol/mg was calculated according to the following formula: MDA = [(ΔA − 0.0445)/0.0166]/Cpr.

### 2.12. Western Blot Analysis

To extract total protein, we scraped cells from Petri dishes with a BeyoGoldTM 21 cm cell scraper (Beyotime, FLFT021, Shanghai, China) and collected them in 1.5 mL centrifuge tubes. After discarding the supernatant, the cells were lysed in RIPA lysis solution (Solarbio, P6730, Beijing, China) containing protease inhibitors (Solarbio, R0020, Beijing, China). After 3 h of lysis at 4 °C, the lysate was centrifuged for 15 min at 5000 rpm. After removing the supernatant, the protein content was measured using the Enhanced BCA Protein Assay Kit (Beyotime, P0010, Shanghai, China). The membranes were then incubated for 12 h at 4 °C with primary antibodies against DHODH (Proteintech, 14877-1-AP, Wuhan, China) and ACTIN (Cell Signaling Technology, 4967S, Boston, MA, USA) using the SDS-PAGE Gel Ultra-Fast Preparation Kit (Beyotime, P0285, Shanghai, China). Following primary antibody incubation, membranes were washed three times (15 min/time) in TBST (Biosharp, BL315B, Hefei, Anhui, China) and incubated for 2 h at room temperature in the dark with hrp-labeled goat anti-rabbit IgG (Beyotime, A0208, Shanghai, China). Membranes were washed three times with TBST and then subjected to chemiluminescence detection (iBrightCL1000, Thermo) on a machine using a BeyoECL Star (Beyotime, P0018AC, Shanghai, China). Image J 1.45s (NIH, Bethesda, MD, USA) was used for image analysis.

### 2.13. Statistical Analysis

A significance level of *p* < 0.05 was considered as the threshold for determining significant differences. Image analysis was performed using Image J 1.45s software (NIH, Bethesda, MD, USA), and GraphPad Prism 8.0 software (GraphPad Software Inc., San Diago, CA, USA) was utilized for graphing purposes. Data were represented by means ± standard deviation (SD).

## 3. Results

### 3.1. Cloning and Characterization of the Coding Region of Yak DHODH Gene

Using agarose gel electrophoresis, we were able to obtain a distinct target band amplification from the coding region of the yak *DHODH* gene (Appendix A). Subsequently, we performed Sanger sequencing to determine the nucleotide sequence of the amplified fragment. The result revealed that the coding region of the yak *DHODH* gene contained 1188 bp, encoding a predicted protein of 395 amino acids (Figure 1). The comparative analysis of the nucleotide sequence similarity of the yak *DHODH* gene revealed a remarkable level of homology with wild yaks and domestic cattle, reaching 99.75%. Additionally, lower degrees of similarity were observed with other species, such as goats (98.23%), camels (95.19%), pigs (95.19%), and horses (93.19%). Moreover, the amino acid homology of the yak *DHODH* gene displayed even lower similarity with tigers (91.65%), humans (91.65%), dogs (90.89%), and rabbits (90.63%) (Appendix A). These findings suggest a significant conservation of the yak *DHODH* gene across diverse species, particularly in cattle.

### 3.2. Physicochemical Properties, Structure, and Functional Analysis of Yak Predicted DHODH Protein

Using ExPASy, we have identified that the predicted molecular weight of the yak DHODH protein was 42,746.02, with a molecular formula of C_1891_H_3083_N_555_O_560_S_6_. Additionally, the protein has an aliphatic index of 98.76, an isoelectric point of 9.47, and an instability index of 39.65. According to Appendix A, lysine (12.9%) and glycine (10.4%) were the most abundant amino acids in the yak DHODH protein. Furthermore, the protein contained 50 positively charged amino acids (Arg and Lys) and 43 negatively charged amino acids (Asp and Glu), resulting in a net positive charge. The average hydropathy index of DHODH was −0.172 (Appendix A), indicating that it possessed stable and lipid-soluble traits, with a hydrophilic nature. In addition, our predictions indicated that yak DHODH was likely to have multiple phosphorylation sites, predominantly involving threonine (Thr), serine (Ser), and tyrosine (Tyr) residues (Appendix A). Furthermore, yak DHODH contained three glycosylation sites and a transmembrane domain (Appendix A). Protein secondary structure prediction revealed that the structure of yak DHODH protein primarily consisted of a random coil and α-helix, the rest of which were β-turns and extended strands (Figure 2A). Protein tertiary structure predictions were consistent with the secondary structure predictions (Figure 2B). In comparison with other species, we found that the amino acid homology of yak DHODH with wild yak (XP_005896557.1), cattle (NP_001015650.1), goat (XP_005692254.2), wild Bactrian camel (XP_006183753.1), pig (XP_020949488.1), horse (XP_014593928.1), tiger (XP_007074994.2), human (NP_001352.2), dog (XP_038394380.1), and rabbit (XP_051702933.1) were 99.75%, 99.75%, 98.23%, 95.19%, 95.19%, 93.16%, 91.65%, 91.65%, 90.89%, and 90.63%, respectively (Figure 2C,D). These results further solidified that the yak DHODH protein demonstrated a remarkable level of conservation across terrestrial animals.

### 3.3. Expression Analysis of the DHODH Gene in Various Tissues of Yaks

We assessed the relative mRNA expression levels of the *DHODH* gene in various tissues of yaks, including ovaries, heart, liver, spleen, lungs, kidneys, uterus, muscles, oviduct, vas deferens, adipose tissue, and epididymis, using RT-qPCR. Our findings revealed a widespread expression of yak *DHODH* mRNA across all examined tissues. Significantly higher expression levels of *DHODH* were observed in the heart and spleen of yaks compared to other tissues (*p* < 0.05). Moreover, the vas deferens exhibited the lowest expression level (Figure 3). These results imply the potential roles of *DHODH* in various tissues of yaks.

### 3.4. Isolation of Yak Skin Fibroblasts (YSFs) and Establishment of Bisphenol S-Induced Toxicity Model

An *in vitro* bisphenol S (BPS)-induced cell model was created utilizing the isolated YSFs. Initially, YSFs were exposed to different concentrations (0, 100, 300, 500, 700, 900, 1100 μM) of bisphenol S (BPS). Subsequently, we assessed the cell morphology and measured the cell viability at different time points (12, 24, and 48 h). Our findings demonstrated that BPS hindered cell growth and triggered cell death in a dose-dependent manner (Figure 4A). Specifically, cell viability sharply decreased with the administration of 700, 900, and 1100 μM BPS (Figure 4B). Furthermore, JC-1 fluorescence staining and subsequent intensity analysis demonstrated a significant elevation in the green-red fluorescence ratio within the 900 and 1100 μM BPS treatment groups (*p* < 0.05) (Figure 4C). This finding suggests that high concentrations of BPS effectively decreased mitochondrial membrane potential, indicating mitochondrial dysfunction. Moreover, ROS detection revealed a significant increase in ROS production within the 700, 900, and 1100 μM BPS groups (*p* < 0.05), indicating the induction of oxidative stress by BPS (Figure 4D). The assessment of glutathione (GSH) content revealed a significant decrease in GSH levels within the 700, 900, and 1100 μM BPS groups (*p* < 0.05), indicating a depletion of cellular antioxidant defenses (Figure 4E). Conversely, the measurement of malondialdehyde (MDA) content in these BPS groups displayed a clear increase (Figure 4F). Additionally, BODIPY fluorescence staining and subsequent statistical analysis demonstrated a significant rise in green fluorescence within the 900 and 1100 μM BPS treatment groups (*p* < 0.05) (Figure 4G), providing strong evidence of lipid peroxidation occurrence in YSFs. In conclusion, we successfully established a BPS-induced toxicity model in YSFs, which showed a reduction in cell viability and GSH and JC-1 levels, and an elevation in ROS level and MDA and BODIPY staining.

### 3.5. Inhibition of Ferroptosis Alleviates BPS-Induced Cellular Toxicity

Furthermore, we conducted additional experiments to verify whether BPS induced ferroptosis in YSFs by co-administering ferroptosis inhibitors (Ferrostatin-1, Fer) with the 900 μM BPS treatment. Through bright-field observation and cell viability analysis, we observed that the addition of 1 μM Fer significantly mitigated BPS-induced cell death and attenuated the reduction in cell viability within 24 h (*p* < 0.05) (Figure 5A,B). Moreover, analysis of JC-1 and ROS fluorescence staining showed that the mitochondrial membrane potential and ROS levels were restored to normal levels in the BPS+Fer-1 group (*p* < 0.05), but not in the group treated with a high dose of Fer (Figure 5C,D). Furthermore, GSH measurements revealed a significant increase in GSH levels when YSFs treated with BPS were supplemented with different doses of Fer (*p* < 0.05) (Figure 5E). Additionally, assessment of MDA and BODIPY levels demonstrated that the addition of 1 μM Fer significantly rescued BPS-induced lipid peroxidation in YSFs (Figure 5F,G). In summary, we confirmed the BPS-induced ferroptosis model in this study by co-administering various doses of ferroptosis inhibitors, and further established that BPS is a potent inducer of ferroptosis.

### 3.6. Overexpression of DHODH Alleviates BPS-Induced Ferroptosis in YSFs

Next, we investigated the potential involvement of the *DHODH* gene in reversing BPS-induced ferroptosis in vitro. To begin, we constructed a vector for overexpressing the yak *DHODH* gene (pcDNA3.1-*DHODH*) and verified its functionality in YSFs by conducting RT-qPCR, Western blot, and fluorescence observation (Figure 6A and Appendix A). Similarly, we assessed the abovementioned markers of ferroptosis, including cell viability, JC-1 staining, ROS levels, GSH levels, MDA levels, and BODIPY staining. Surprisingly, the overexpression of the *DHODH* gene resulted in enhanced cell growth and significantly counteracted the inhibitory effect of BPS on cell viability in YSFs (*p* < 0.05) (Figure 6B,C). Additionally, the overexpression of the *DHODH* gene attenuated the reduction in mitochondrial membrane potential and the increase in ROS levels induced by BPS treatment in YSFs (Figure 6D,E). Interestingly, the overexpression of the *DHODH* gene demonstrated antioxidant activity, as evidenced by the elevation of GSH levels and reduction in MDA levels in YSFs (*p* < 0.05) (Figure 6F,G). Moreover, the levels of GSH and MDA in BPS-cultured YSFs were restored to normal levels upon overexpression of the *DHODH* gene (*p* < 0.05) (Figure 6F,G). Additionally, fluorescence staining and statistical analysis indicated a significant reduction in BODIPY intensity in the OE+B group (BPS 900 μM + overexpressing DHODH) compared to the NC+B group (BPS 900 μM + negative control) (*p* < 0.05), further supporting the role of the *DHODH* gene in counteracting BPS-induced ferroptosis (Figure 6H). In conclusion, our study offers valuable insights into the inhibitory effect of *DHODH* on BPS-induced ferroptosis in YSFs.

### 3.7. siRNA Interference of DHODH Exacerbates BPS-Induced Ferroptosis in YSFs

To further investigate the role of *DHODH* in YSFs’ ferroptosis, we employed RNA interference (RNAi) to suppress *DHODH* expression. Firstly, we confirmed the efficiency of the siRNA used in our study for knocking down *DHODH* (Figure 7A). Bright-field observations and cell viability assays demonstrated that, compared to group NC+B (negative control + 900 μM BPS), interfering with the *DHODH* gene in BPS-treated YSFs exacerbated the suppression of cell growth and viability (*p* < 0.05) (Figure 7B,C). In comparison to the NC+B group, *DHODH* gene interference resulted in a reduction in mitochondrial membrane potential (JC-1 staining) in the siRNA+B group (siRNA interference + 900 μM BPS) (*p* < 0.05) (Figure 7D). Furthermore, ROS fluorescence imaging and statistical analysis revealed a significant increase in green fluorescence intensity indicative of ROS in the siRNA+B group compared to the NC+B group (*p* < 0.05) (Figure 7E). Similarly, the statistical analysis for GSH showed a significant decrease in GSH levels in the siRNA+B group compared to the NC+B group (*p* < 0.05) (Figure 7F). Additionally, the results for MDA indicated a notable increase in MDA levels in the siRNA+B group compared to the NC+B group (*p* < 0.05) (Figure 7G). Finally, analysis of BODIPY fluorescence images and statistical analysis revealed a significant increase in BODIPY green fluorescence in the siRNA+B group compared to the NC+B group (*p* < 0.05) (Figure 7H). In conclusion, through the use of RNA interference, we have confirmed the inhibitory role of the yak *DHODH* gene in ferroptosis of BPS-treated YSFs, providing valuable insights into the regulatory mechanisms of *DHODH* in the ferroptosis of yak cells.

## 4. Discussion

The yak is predominantly found in the Himalayas and its surrounding regions in Central Asia. Yaks serve as vital assets for the livelihoods and sustenance of local farmers and herders, playing a crucial role in facilitating regional economic growth and maintaining social stability [27]. Given the challenging environment of the high-altitude plateau, marked by severe cold, intense ultraviolet radiation, low oxygen levels, and limited food resources, yaks often face various forms of stress. These stressors can give rise to several challenges that have a significant impact on yak production and reproductive capabilities [28]. Adverse stressors can induce PCD, such as ferroptosis, a process in which the *DHODH* gene has been identified to play a regulatory role in experimental animal models [29]. Exploring the coding region of the yak *DHODH* gene holds significance in understanding its resilience, thereby enhancing yak survival and reproductive capacity in the face of challenging conditions.

Consequently, in the present study, the cloning and characterization of the yak *DHODH* gene were carried out to thoroughly investigate its coding region structure and homology, marking the first systematic exploration in this regard. The PCR amplification of the yak *DHODH* gene yielded successful cloning outcomes, with distinct target bands being obtained. Subsequent sequencing analysis unveiled that the coding region of the yak *DHODH* gene spans 1188 bp, encoding a protein composed of 395 amino acids. Post-translational modifications of the protein, including phosphorylation, glycosylation, acetylation, methylation, etc., directly regulate its properties and functions, to some extent determining the normal function of the protein [30]. Furthermore, through a comparative analysis of the amino acid sequences of *DHODH* genes in different animal species, we observed that the yak *DHODH* gene exhibited the highest level of homology, at 99.75%, with wild yaks and domestic cattle, implying a remarkable conservation of the yak *DHODH* gene among these species. Notably, Sierra et al. [31] also reported a similar high conservation of *DHODH* between mammals and Toxoplasma gondii, which aligns with our findings.

DHODH plays a crucial role in the purine nucleotide synthesis pathway, and its physicochemical properties are essential for investigating its functions and involvement in ferroptosis [32]. Hence, we utilized ExPASy to predict the molecular weight, molecular formula, aliphatic index, isoelectric point, instability index, and other pertinent information of the predicted yak DHODH protein. Our findings showcased that lysine and glycine were the most abundant amino acids, indicating a hydrophilic nature of the protein. This finding is consistent with the work of Sven et al. [33], who previously demonstrated the hydrophilic properties of DHODH. Moreover, we predicted the secondary and tertiary structures, which revealed a predominance of random coils and α-helices in the DHODH protein. This aligns with the conclusions derived by Lin et al. [34] and others, who found a similar composition of random coils and α-helices in the Leishmania *DHODH* protein.

BPS, a substitute for BPA, is commonly found in cosmetics, food processing, and various other products. Studies have indicated that BPA may have the potential to cause endocrine disruption and toxicity in animals) [35]. To assess the toxic effects of BPS on yak, we conducted an *in vitro* study using YSFs as a model. Our investigation focused on examining the impact of BPS on cell growth, oxidative stress, and lipid peroxidation. The findings showed a dose-dependent decrease in cell growth, cell viability, and mitochondrial membrane potential in YSFs when exposed to increasing concentrations of BPS. Additionally, there was a significant increase in oxidative stress levels and lipid peroxidation. These results are consistent with Mimi et al.’s [36] research on oocyte and sperm cells, which reported inhibited cell growth, heightened oxidative stress, and cellular lipid peroxidation caused by BPS exposure.

Current research suggests that BPS might impact cellular health by triggering oxidative stress and lipid peroxidation, both of which were the indicators of ferroptosis [37]. Additionally, Kaptaner et al. [38] suggested that BPS leads to oxidative stress and lipid peroxidation in rainbow trout liver cell. More strikingly, a study by Bao et al. [39] indicated that PBA could induce ferroptosis in renal tubular epithelial cells. In the present study, we made a significant observation that the introduction of ferroptosis inhibitors effectively mitigated the cellular harm induced by BPS, thereby preserving cell growth and viability. Remarkably, this finding is consistent with the conclusions drawn by Liu et al. [40], who observed similar outcomes in the human bronchial epithelial cell line BEAS-2B when treated with a ferroptosis inhibitor called Fer-1. Consequently, our results suggest that excessive exposure to BPS can induce ferroptosis, while the administration of ferroptosis inhibitors proves to be an effective preventative measure against this process.

Increasing evidences indicate a certain connection between *DHODH* and ferroptosis [41]. Results from Li et al. [42] point out that *DHODH* regulated neuronal ferroptosis following spinal cord injury. As a novel form of cell death, ferroptosis is associated with the accumulation of intracellular free iron and the process, leading to oxidative stress [43]. *DHODH*, being an essential constituent of the mitochondrial respiratory chain, plays a critical role in maintaining cellular function. However, any disruptions in its normal activity can cause a decline in mitochondrial membrane potential and an excessive generation of reactive oxygen species (ROS), thereby exacerbating oxidative stress and potentially instigating ferroptosis [44]. Therefore, according to experiments involving *DHODH* overexpression and siRNA interference, we further elucidated the regulatory role of *DHODH* in regulating the ferroptosis of yak YFSs. We found that overexpression of *DHODH* significantly enhances cell growth, reduces oxidative stress levels, and maintains mitochondrial membrane potential stability. Conversely, interference with *DHODH* leads to reverse effects. Our siRNA interference experiments corroborated the observations made by Liu et al. [45], who discovered that the inhibition of *DHODH* leads to cell cycle arrest and cell death in melanoma cells. Moreover, our overexpression experiments were in agreement with the findings reported by Qian et al. [46], highlighting the role of increased *DHODH* expression in enhancing cell proliferation, which was further substantiated in our own investigation. Thus, we have provided further evidences to support the essential and intricate roles of yak *DHODH* in regulating cell metabolism, responding to oxidative stress, and ensuring cell survival.

In conclusion, this study represents the first cloning of the coding region of the yak *DHODH* gene, allowing us to characterize its biochemical properties. These findings will contribute significantly to the further elucidation of its functional roles in various biological processes. We further confirmed the basic role of *DHODH* gene in anti-ferroptosis of yak YSFs via gene overexpression and siRNA interference. Those findings not only enhance our understanding of the mechanisms behind ferroptosis but also shed light on comprehending the survival strategies of yaks in extreme environments. Particularly, in terms of reproductive adaptability, *DHODH* may play a pivotal role in responses of yak to external pressures and adaptive regulation. Thus, the results provide potential insights for the protection and maintenance of the yak.

## 5. Conclusions

The coding region of the yak *DHODH* gene spanned 1188 bp, encoding a protein consisting of 395 amino acids. The yak *DHODH* gene exhibited a high degree of conservation across mammalian species and was widely expressed in yaks. Through the overexpression and interference of the yak *DHODH* gene in BPS-induced ferroptosis of YSFs, we confirmed its role in resisting ferroptosis, offering a fresh and unique perspective on the mechanisms behind *DHODH* in the stress responses of yaks.

## Figures and Tables

**Figure 1 animals-13-03832-f001:**
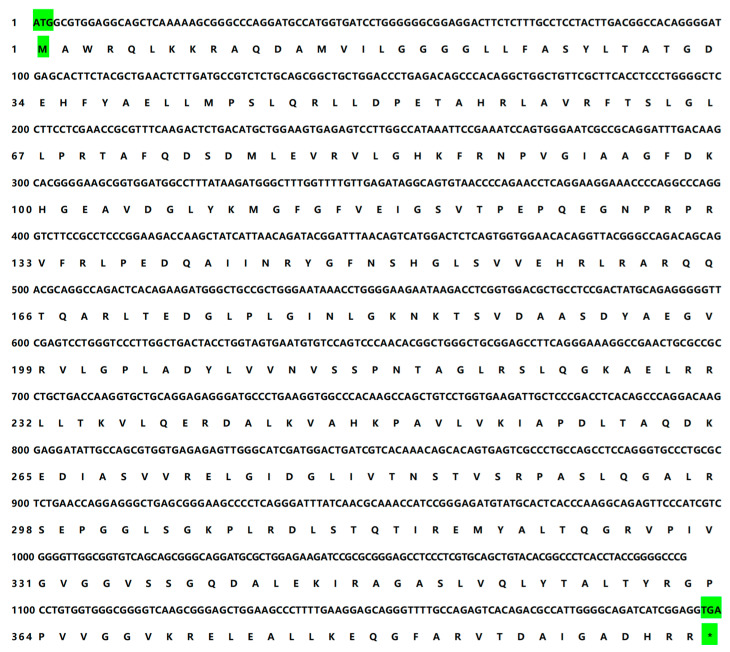
Cloning and characterization of coding region of the yak *DHODH* gene. The translation start codon is highlighted with a green background. The translation stop codon is highlighted with a green background and an asterisk (*).

**Figure 2 animals-13-03832-f002:**
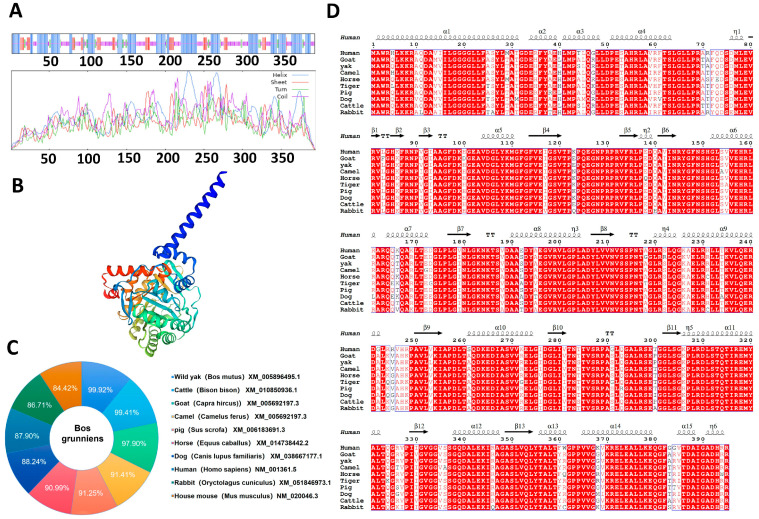
Physicochemical properties, structure, and functional prediction of yak predicted DHODH protein. (**A**) Secondary structure prediction of the yak DHODH protein highlighting α-helices in blue, extended strands in red, β-turns in green, and random coils in purple. (**B**) Tertiary structure prediction of the yak DHODH protein. (**C**) Comparison of amino acid homology of DHODH between yaks and various species. (**D**) Conserved and similar residues among species highlighted with red and white boxes, respectively.

**Figure 3 animals-13-03832-f003:**
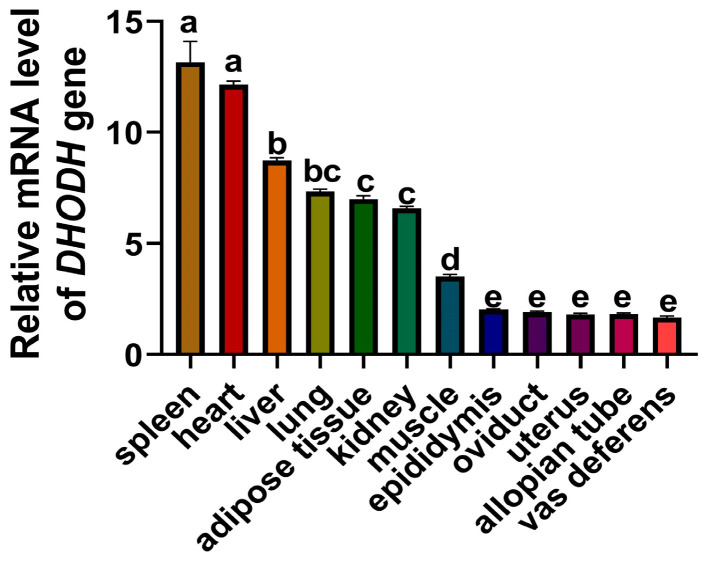
Expression analysis of *DHODH* gene in various tissues of yaks. Different letters upon panels mean significant differences, *n* ≥ 3.

**Figure 4 animals-13-03832-f004:**
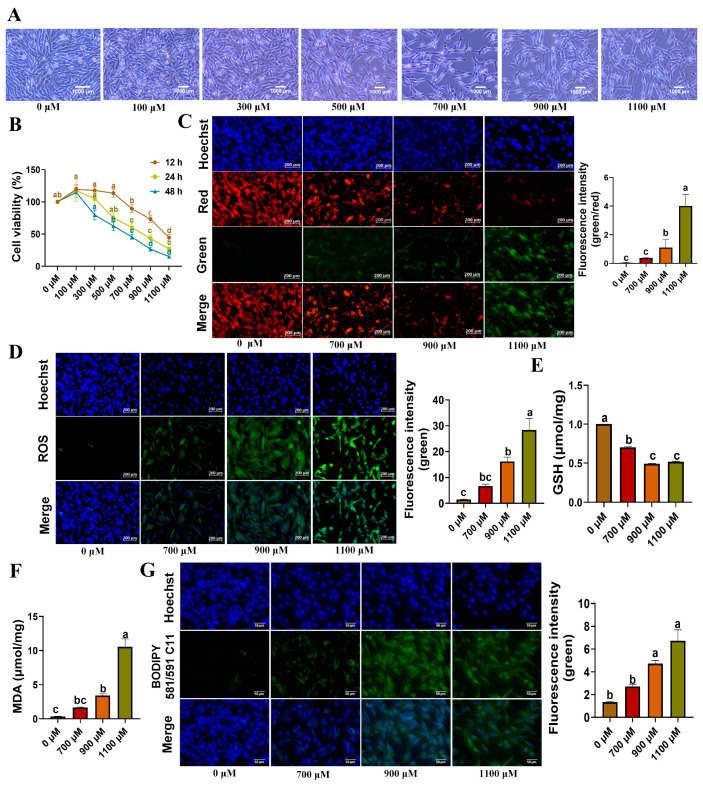
Establishment of the BPS-induced toxicity model. (**A**) Bright-field photographs of YSFs treated with 0–1100 μM BPS for 24 h. (**B**) Cell viability of YSFs treated with 0–1100 μM BPS for 12, 24, and 48 h time intervals, *n* = 3. (**C**) Mitochondrial membrane potential detection under fluorescence microscopy in YSFs treated with 0, 700, 900, and 1100 μM PBS, scale bar = 200 μm; fluorescence values and green-to-red fluorescence ratio quantification performed using Image J 1.45s software, *n* ≥ 3. (**D**) ROS fluorescence images observed under fluorescence microscopy in YSFs treated with 0, 700, 900, and 1100 μM PBS, scale bar = 200 μm; fluorescence values quantified using Image J software, *n* = 3. (**E**) GSH content in YSFs treated with 0, 700, 900, and 1100 μM BPS for 24 h, *n* = 3. (**F**) MDA content in YSFs treated with 0, 700, 900, and 1100 μM BPS for 24 h, *n* = 3. (**G**) BODIPY fluorescence images in YSFs treated with 0, 700, 900, and 1100 μM BPS for 24 h, scale bar = 50 μm; green fluorescence values quantified using Image J 1.45s, *n* = 3. Different letters upon panels mean significant differences.

**Figure 5 animals-13-03832-f005:**
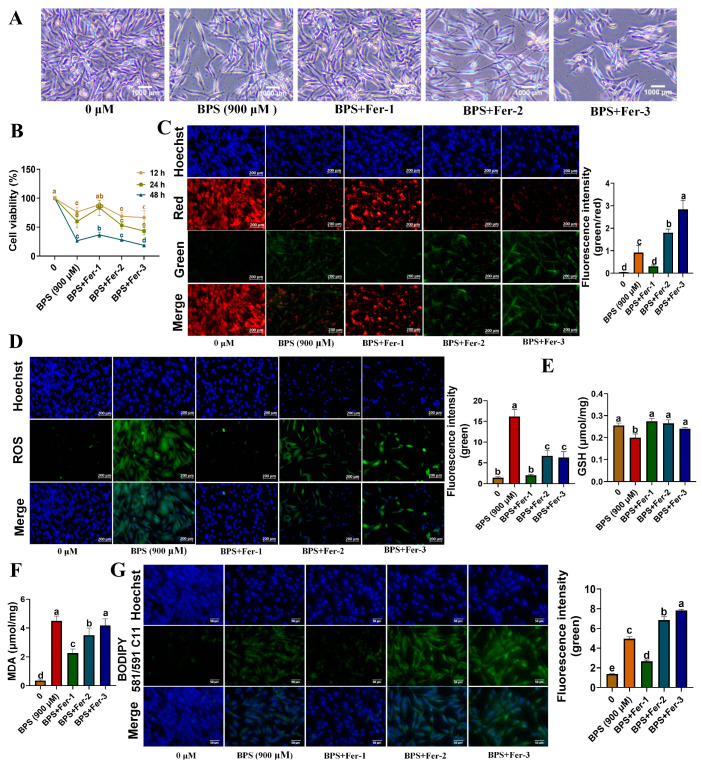
Verification of ferroptosis in YSFs induced by BPS. (**A**) Bright-field photographs of YSFs in control (0 μM), BPS 900 μM (BPS), BPS 900 μM with 1 μM Fer (BPS+Fer-1), BPS 900 μM with 10 μM Fer (BPS+Fer-2), and BPS 900 μM with 100 μM Fer (BPS+Fer-3) treatments for 24 h. (**B**) Cell viability of YSFs treated with BPS and different concentrations of Fer for 12, 24, and 48 h time intervals, *n* = 3. (**C**) Mitochondrial membrane potential observed under fluorescence microscopy in YSFs treated with different combinations of BPS and Fer for 24 h, scale bar = 200 μm; Fluorescence values and green-to-red fluorescence ratio quantified using Image J 1.45s, *n* ≥ 3. (**D**) ROS fluorescence images determined under fluorescence microscopy in YSFs treated with different combinations of BPS and Fer for 24 h, scale bar = 200 μm; Fluorescence values quantified using Image J 1.45s, *n* = 3. (**E**) GSH content in YSFs treated with different combinations of BPS and Fer for 24 h, *n* = 3. (**F**) MDA content in YSFs treated with different combinations of BPS and Fer for 24 h, *n* = 3. (**G**) BODIPY fluorescence images in YSFs treated with different combinations of BPS and Fer for 24 h, scale bar = 50 μm; Green fluorescence values quantified using Image J 1.45s, *n* = 3. Different letters upon panels meant significant difference.

**Figure 6 animals-13-03832-f006:**
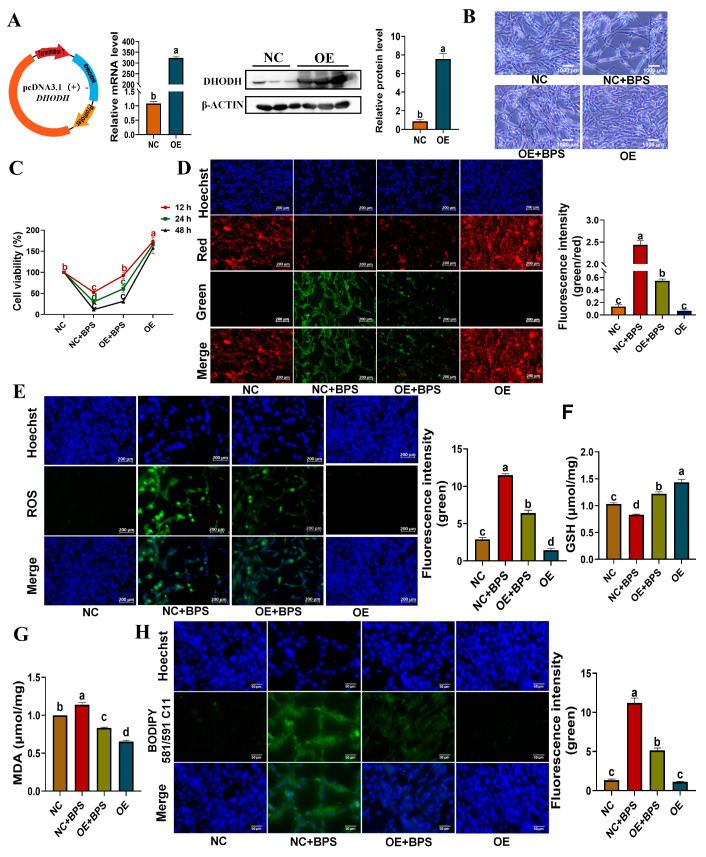
Validating the protective effect of *DHODH* on YSFs. (**A**) *DHODH* plasmid construction and verification of *DHODH* overexpression in YSFs in the negative control (NC) and *DHODH* overexpression (OE) groups via RT-qPCR and Western blot, *n* ≥ 3. (**B**) Bright-field photographs of YSFs treated with negative control plasmid (NC), BPS 900 μM + negative control plasmid (NC+B), BPS 900 μM + *DHODH* overexpression plasmid (OE+B), and *DHODH* overexpression plasmid (OE) for 24 h. (**C**) Cell viability of YSFs treated with different conditions for 12, 24, and 48 h time intervals, *n* = 3. (**D**) Mitochondrial membrane potential changes observed under fluorescence microscopy in YSFs treated with different conditions for 24 h, scale bar = 200 μm; Fluorescence values and green-to-red fluorescence ratio quantified using Image J 1.45s, *n* = 3. (**E**) ROS fluorescence images observed under fluorescence microscopy in YSFs treated with different conditions for 24 h, scale bar = 200 μm; Fluorescence values quantified using Image J 1.45s, *n* = 3. (**F**) GSH content in YSFs treated with different conditions for 24 h, *n* = 3. (**G**) MDA content in YSFs treated with different conditions for 24 h, *n* ≥ 3. (**H**) BODIPY fluorescence images in YSFs treated with different conditions for 24 h, scale bar = 50 μm; Green fluorescence values quantified using Image J 1.45s, *n* ≥ 3. Different letters upon panels mean significant differences.

**Figure 7 animals-13-03832-f007:**
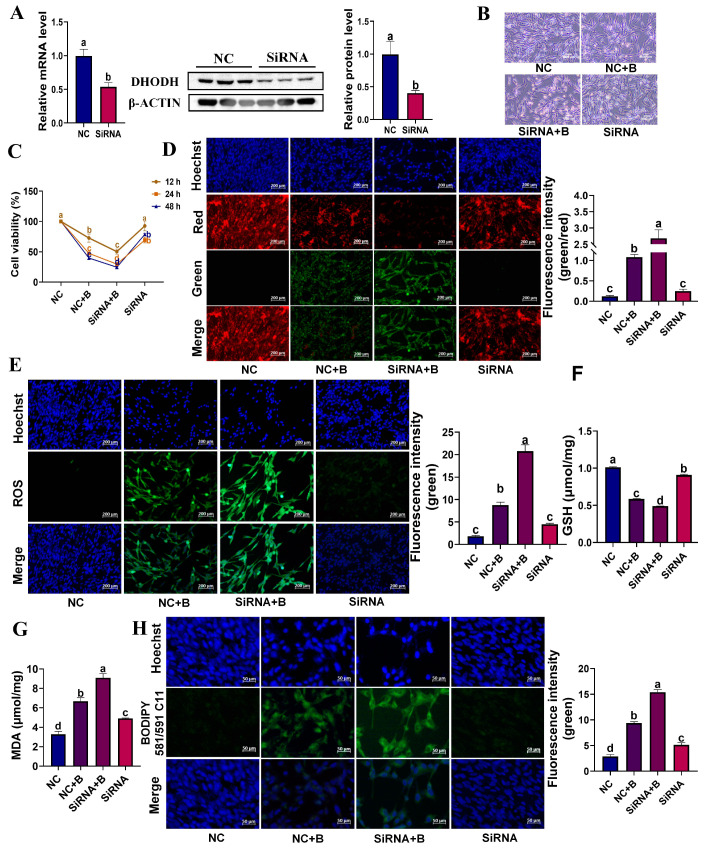
The exacerbating effects of siRNA interference of *DHODH* gene on BPS-induced ferroptosis in YSFs. (**A**) RT-qPCR and Western blot validation of DHODH interference in control (NC) and the small interfering DHODH group (siRNA), *n* ≥ 3. (**B**) Bright-field images of YSFs treated for 24 h in the control (NC), control + 900 μM BPS (NC+B), siRNA interference + 900 μM BPS (siRNA+B), and siRNA interference (siRNA) groups. (**C**) Cell viability of YSFs treated for 12, 24, and 48 h in different groups, *n* = 3. (**D**) Mitochondrial membrane potential changes observed under a fluorescence microscope after 24 h in different treatment groups, scale = 200 μm; Image analysis of green–red fluorescence ratios performed using Image J 1.45s, *n* ≥ 3. (**E**) ROS fluorescence images of YSFs under different treatments, scale = 200 μm; Image analysis of fluorescence values preformed using Image J 1.45s, *n* = 3. (**F**) GSH content in yak fibroblasts after 24 h in various treatment groups, *n* = 3. (**G**) MDA content in YSFs after 24 h in different groups, *n* ≥ 3. (**H**) BODIPY fluorescence images of YSFs after 24 h in different treatment groups, scale = 50 μm; Image analysis of green fluorescence values using Image J 1.45s, *n* = 3. Different letters upon panels meant significant differences.

## Data Availability

The data presented in this study are available in article and Appendix A.

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
