# Peer review of "Cloning and Characterization of Yak DHODH Gene and Its Functional Studies in a Bisphenol S-Induced Ferroptosis Model of Fetal Fibroblasts"

_animals, 2023, doi:10.3390/ani13243832_

Round 1

Reviewer 1 Report

Comments and Suggestions for Authors

This manuscript provided compelling evidence for the essential role of yak DHODH in the regulation of ferroptosis. Through the cloning and characterization analysis of the DHODH gene in yaks, a BPS-induced feroptosis model of yak YSFs was studied. They provided a fresh and unique perspective on the mechanisms behind DHODH in the stress responses of yaks. The results is formative and well presented. 

Several suggestions:

1. The abstract is too long, should be more highlighted.

2. The item 3.3, their findings revealed a wide spread expression of yak DHODH mRNA across all examined tissues,and the expressional levels were significant different. This point is good. However, they suggested that the critical roles of DHODH in various life processes of yaks based the above point. This is not logical and not enough evident.

Author Response

Response to Reviewer 1 Comments

1. Summary

2. Questions for General Evaluation

Reviewer’s Evaluation

Response and Revisions

Does the introduction provide sufficient background and include all relevant references?

Yes

Are all the cited references relevant to the research?

Yes

Is the research design appropriate?

Yes

Are the methods adequately described?

Yes

Are the results clearly presented?

Yes

Are the conclusions supported by the results?

Yes

3. Point-by-point response to Comments and Suggestions for Authors

Comments 1: The abstract is too long, should be more highlighted.

Response 1: Thank you for pointing this out. We agree with this comment. In our revised version, we updated and simplified the words in the section of Abstract, please find them in Line 24-49.

Comments 2: The item 3.3, their findings revealed a wide spread expression of yak DHODH mRNA across all examined tissues, and the expressional levels were significant different. This point is good. However, they suggested that the critical roles of DHODH in various life processes of yaks based the above point. This is not logical and not enough evident.

Response 2: We agree with this reviewer’s comment. We modified our description in our resubmitted manuscript. (Please see the Line 351)

Reviewer 2 Report

Comments and Suggestions for Authors

The authors cloned the yak DHODH gene and primarily explore the mechanism of its function in ferroptosis. But the experimental design is not complete. Several questions listed as follows.

Q1:How did the authors distinguish the ferroptosis and apoptosis?

Q2: The beginning sentence in line 348 did not match the paragraph content.

Q3: Many pictures in the article are not clear. For example the merge and green field of 1100um in figure 3C seems the same and the merge and green field of 900um in figure 3D are blurry. Similar problems also occurred in Figure 4,5 and 6.

Q4: The authors didn’t confirm the protein expression of DHODH gene in the control and overexpression level.

Q5: Why didn’t the authors consider ferroptosis inhibitors, BPS and DHODH treat together?

Author Response

Response to Reviewer 2 Comments

1. Summary

2. Questions for General Evaluation

Reviewer’s Evaluation

Response and Revisions

Does the introduction provide sufficient background and include all relevant references?

Yes

Are all the cited references relevant to the research?

Yes

Is the research design appropriate?

Must be improved

Thank you for your comment, changes have been made.

Are the methods adequately described?

Can be improved

Thank you for your comment, changes have been made.

Are the results clearly presented?

Must be improved

Thank you for your comment, changes have been made.

Are the conclusions supported by the results?

Must be improved

Thank you for your comment, changes have been made.

3. Point-by-point response to Comments and Suggestions for Authors

Comments 1: How did the authors distinguish the ferroptosis and apoptosis?

Response 1: Thank you for pointing this out. Indeed, there are similar phenotypes of ferroptosis and apoptosis, such as the increase of ROS and cell death, and the decrease of cell viability, JC-1, and GSH levels. In this experiment, we specifically detected two kinds of ferroptosis indicators, MDA and BODIPY staining, both of which were significantly upregulated in BPS-cultured yak cells, implying the occurrence of ferroptosis. More importantly, to verify this result, three dosages of ferroptosis inhibitor, Ferrostatin-1 (Fer; 1, 10, and 100 μM), were used to co-treat BPS-cultured yak cells, and we discovered that 1 μM Fer could rescue lipid peroxidation, cell viability, and death in by recovering CCK-8, JC-1, ROS, GSH, MDA, and BODIPY to normal level. According to the abovementioned evidences, we successfully confirmed the ferroptosis inducer role of BPS in yak cells and distinguished the processes of ferroptosis and apoptosis.

Comments 2: The beginning sentence in line 348 did not match the paragraph content.

Response 2: Thanks for this reviewer’s question. We have therefore revised this section and, to emphasize this point, the text from 358-359 is highlighted in yellow.

Comments 3: Many pictures in the article are not clear. For example, the merge and green field of 1100 μm in Figure 3C seems the same and the merge and green field of 900 μm in Figure 3D are blurry. Similar problems also occurred in Figures 4,5 and 6.

Response 3: Thank you for pointing this out. We checked the graphs and replaced them with our original pictures. However, when we uploaded our manuscript for the first, all pictures were high-definition. Therefore, to avoid these mistakes, we provided the PDF version of all figures in our resubmitted version, please find them in attachments.

Comments 4: The authors didn’t confirm the protein expression of the DHODH gene in the control and overexpression levels.

Response 4: Thank you for your insightful comments. We conducted the overexpression and interference experiments, and completed the results in Figure 6A and Figure 7A, respectively. All the results were in line with our expectations. The additional explanations were perfected in the revised version, please find them in Line 442-444 and 477-479 in the section of Results, and Line 271-287 in the section of Materials and Methods.

Comments 5: Why didn’t the authors consider ferroptosis inhibitors, BPS, and DHODH treat together?

Response 5: We thank you for this review’s insightful suggestions. In our study, we used BPS to induce an in vitro ferroptosis model in yak cells and confirmed the induced role of BPS in ferroptosis. To validate this result, we added ferroptosis inhibitor, Fer, and further verified our cell model of ferroptosis. Next, to explore the anti-ferroptosis role of DHODH in yak cells, we overexpressed and interfered with it and found that DHOHD exerted anti-ferroptosis in yaks. Over all, we achieved our goals for exploring the anti-ferroptosis role of DHODH in yak cells. We completely agree with the reviewer’s suggestions, and we will try the combination of Fer, BPS, and DHOHD to treat yak cells in our next experiment.

Reviewer 3 Report

Comments and Suggestions for Authors

Dear Authors,

the manuscript proposal „Cloning and Characterization of Yak DHODH Gene and Its Functional Studies in a bisphenol S-Induced Ferroptosis Model of Fetal Fibroblasts“ is interesting.

The Introduction chapter is clear and provides a good basis for understanding the research topics.

In the Materials and Methods chapter, the research is clearly described.

The results are clearly presented. In the Discussion chapter, you clearly place the research results in the context of early scientific research.

The conclusion is clear and concise. Perhaps the conclusion could be expanded slightly.

Minor technical errors were noted in the citation of references in the text of the Discussion chapter and in the citation of references at the end of the manuscript.

Edit the references in lines 507, 516, 519, 529, 534, 535, 539, 545, 558, 560 and write the reference number [xx] in brackets after et al.

The references in the bibliography should be edited according to the instructions of the journal Animals.

Author Response

Response to Reviewer 3 Comments

1. Summary

2. Questions for General Evaluation

Reviewer’s Evaluation

Response and Revisions

Does the introduction provide sufficient background and include all relevant references?

Yes

Are all the cited references relevant to the research?

Yes

Is the research design appropriate?

Yes

Are the methods adequately described?

Yes

Are the results clearly presented?

Yes

Are the conclusions supported by the results?

Yes

3. Point-by-point response to Comments and Suggestions for Authors

Comments 1 The conclusion is clear and concise. Perhaps the conclusion could be expanded slightly.

Response 1: Thank you for this kindly suggest. In our revised manuscript, we expanded the Conclusion to fulfil the reviewer’s requirement. (please see Line 584-589)

Comments 2: Minor technical errors were noted in the citation of references in the text of the Discussion chapter and in the citation of references at the end of the manuscript.

Edit the references in lines 507, 516, 519, 529, 534, 535, 539, 545, 558, 560 and write the reference number [xx] in brackets after et al.

The references in the bibliography should be edited according to the instructions of the journal Animals.

Response 2: Thank you for pointing this out. We agree with this comment, and I have corrected them. (Please see Line 514, 523, 526, 536, 541, 542, 546, 552, 565, and 567).